# Paradoxical Behavior of Oncogenes Undermines the Somatic Mutation Theory

**DOI:** 10.3390/biom12050662

**Published:** 2022-04-30

**Authors:** Noemi Monti, Roberto Verna, Aurora Piombarolo, Alessandro Querqui, Mariano Bizzarri, Valeria Fedeli

**Affiliations:** Systems Biology Group Lab, Department of Experimental Medicine, “Sapienza” University of Rome, Viale Regina Elena 324, 00161 Rome, Italy; noemi.monti@uniroma1.it (N.M.); roberto.verna@fondazione.uniroma1.it (R.V.); aurora.piombarolo@gmail.com (A.P.); alessandroquerqui@gmail.com (A.Q.); mariano.bizzarri@uniroma1.it (M.B.)

**Keywords:** tumor microenvironment, TOFT, paradoxes, cancer reversion, SMT, systems biology

## Abstract

The currently accepted theory on the influence of DNA mutations on carcinogenesis (the Somatic Mutation Theory, SMT) is facing an increasing number of controversial results that undermine the explanatory power of mutated genes considered as “causative” factors. Intriguing results have demonstrated that several critical genes may act differently, as oncogenes or tumor suppressors, while phenotypic reversion of cancerous cells/tissues can be achieved by modifying the microenvironment, the mutations they are carrying notwithstanding. Furthermore, a high burden of mutations has been identified in many non-cancerous tissues without any apparent pathological consequence. All things considered, a relevant body of unexplained inconsistencies calls for an in depth rewiring of our theoretical models. Ignoring these paradoxes is no longer sustainable. By avoiding these conundrums, the scientific community will deprive itself of the opportunity to achieve real progress in this important biomedical field. To remedy this situation, we need to embrace new theoretical perspectives, taking the cell–microenvironment interplay as the privileged pathogenetic level of observation, and by assuming new explanatory models based on truly different premises. New theoretical frameworks dawned in the last two decades principally focus on the complex interaction between cells and their microenvironment, which is thought to be the critical level from which carcinogenesis arises. Indeed, both molecular and biophysical components of the stroma can dramatically drive cell fate commitment and cell outcome in opposite directions, even in the presence of the same stimulus. Therefore, such a novel approach can help in solving apparently inextricable paradoxes that are increasingly observed in cancer biology.

## 1. Introduction

A paradox is a “self-contradictory statement or a finding that runs contrary to one’s expectation”, and it leads to unacceptable conclusions. Here, the word “unacceptable” refers to the theoretical assumptions upon which the experiment has been planned. In other words, a paradoxical finding is a *true* phenomenon than cannot accommodate—or even contradicts—the theoretical premises that should explain the process under scrutiny.

Cancer is usually portrayed—in many scientific papers, textbooks, and even in popular culture—as a “genetic disease” [1], a pathological state arising from “genetic changes”—the “somatic mutations”—that lead to “new” phenotypic traits [2]. Eventually, these features enable cells to recover full motility, invasiveness, and metastatic capability. Yet, it is quite disturbing that such a theory—the Somatic Mutation Theory of Carcinogenesis (SMT)—is deemed a “fact”, i.e., an undisputed and, even worse, an unquestionable fact that has paved the way of cancer research for the last 50 years. The SMT posits that cancer emerges from deregulated/mutated gene expression patterns, starting at the cellular level, from a single “renegade” cell in which the intricate balance between oncogenes and tumor repressor genes is impaired. This framework fostered a bewildering advance in molecular-based technologies. Unfortunately, high-throughput technologies accumulated a large body of data that, in several cases, openly contradicted the SMT’s basic premises, mostly based upon a linear relationship between “signaling molecules” (conceived as “causes”) and their downstream consequences (“effects”) [3,4].

This evidence is further reinforced by the clinical inadequacy of current target-based treatments. Indeed, despite some progress, conventional as well as innovative treatments did not obtain a significant improvement in overall survival, as the overall cancer death rates have remained substantially unchanged or show only slight reduction [5]. As already highlighted [6], these results are far from those expected when we consider the relevant financial commitment currently requested by the pharmaceutical companies.

These disappointing results represent an unavoidable sign of the inadequacy of our theoretical models, and ask for a reappraisal of the conceptual and methodological basis on which cancer research is rooted. This sounds quite reasonable; however, both institutions and individual scientists find it more appealing to continue working along canonical landscapes instead of investigating unchartered alternatives [7].

Overall, the above considerations suggest redesigning clinical and pharmacological studies according to different premises and new cancer paradigms. While studies based on animals are still essential in the field of cancer research, experimental investigations carried out on a specific set of animal models—including inbred, tumor-prone, tumor-cell-injected, or knockout rodent models—carry a number of shortcomings and, in some cases, are mere artifacts. For instance, it is widely recognized that mice—besides being the most commonly animal used—are actually poor models. Indeed, for more than 4000 genes, transcription factor binding sites differ between mice and humans in 41% to 89% of cases [8]. In many cases, mouse models serve to replicate specific processes associated with the disease, but not the whole spectrum of physiological changes in the organism. As a result, it is quite disappointing that the average rate of successful translation from animal models to clinical cancer trials is less than 8% [9].

However, we should ask ourselves if these models are pertinent to the question under scrutiny. Mostly, models are designed to support the accepted theoretical assumptions, not for exploring divergent hypothesis or infer new perspectives [10]. However, even in this case, it is how we are willing to interpret the results that would provide fruitful insights. This claim suggests that even older experiments using animal models can be reread and reinterpreted according to a different perspective. For example, a recent paper [11] indicated that extracellular signals from tissues can be a determining factor in whether oncogenes drive skin cells towards or away from cancer. Another study using mice suggested that growth arrest in melanocytes cannot be an entirely cell-intrinsic process, but must involve extrinsic signals, such as growth inhibitory proteins secreted by neighboring cells [12]. Similarly, it has been observed that claudins can be either tumor promoters or tumor suppressors depending on the context, which emphasizes the importance of the microenvironment when studying the biological functions of so-called oncogenes [13]. In these cases, extrinsic, microenvironment-triggered cues can “manipulate” oncogene activity, thus committing cells toward a very different fate [14].

Noticeably, even followers of the canonical carcinogenesis framework have acknowledged that a number of contradictions weaken the SMT paradigm. Accordingly, “half a century of cancer research had generated an enormous body of observations [...] but there were essentially no insights into how the disease begins and progresses” [15]. Despite the expectations raised by the Ames’ axiom (“substances act as carcinogens because they have mutagenic activity”), it shortly turned out that most powerful carcinogens are actually not mutagenic; “but fortunately—as Weinberg candidly admits—I and others were not derailed by discrepant facts”. We strongly posit that these “discrepant facts” deserve now to be appropriately investigated and explained. This attempt would probably lead to a profound reframing of the pathogenic basis of cancer research, looking at the right level of observation, eventually by reinterpreting accumulated data at the light of a different perspective [16].

## 2. Paradoxes

Reconsidering paradoxes resulting from a wide array of experiments and clinical observations may shed light on this controversy. However, we have to start by reinterpreting those paradoxes that are explicitly left aside by the vast majority of studies, just because the prevailing paradigm—the SMT hypothesis—might not accommodate these findings [17]. Yet, such paradoxical results disclose new perspectives in investigation, and provide support to different theoretical models of carcinogenesis, such as that described in the Tissue Organization Field Theory (TOFT) [18]. TOFT posits that cancer arises from the deregulated dynamical interplay among cells and the tumor microenvironment (TME. Consequently, cancer should be viewed as a tissue-based disease that sustains proliferation as the default state of all cells. The TME is currently accepted as the context—the morphogenetic field—in which a complex array of biophysical and molecular cues participate in shaping cell phenotypes travelling across different “attractors”—i.e., stable functional configurations of a cell’s activity—within the Waddington’s landscape [19]. As such, the TME displays the unique features of a field, acting through non-linear effects and long-range correlations [20].

SMT advocates that ad hoc hypotheses could explain a number of contradictions gathered so far. These “corrections” have been integrated as new “hallmarks” into the classical SMT model. However, they resemble the epicycles of yore, by analogy with the Ptolemy–Copernicus controversy [21]. Unfortunately, new added hypotheses trigger further paradoxes and contradictions, thus requiring additional adjustments that, ultimately, weaken the SMT further. Definitely, a different theoretical perspective is urgently needed to face such conundrums [22].

### 2.1. Mutations

Cancer cells display a wide array of somatic mutations. Noticeably, the pattern of gene mutations shows remarkable intratumor differences as well as significant changes over time, from the early stage of tumor development up to the spreading of distant metastases. These features notwithstanding, mutations have been interpreted in causative terms: the gain or loss of function due to mutation(s) is thought to trigger specific modifications in cell functions [23]. However, the demonstration of a mechanistic, truly causative link between oncogenes and cancer is still lacking, and the supposed correlation is suggestive of *association and not causation*. As clearly stated by Woodman and Mills [24] “the spectrum of events that must occur in a single cell, including which oncogene activation and tumor suppressor gene inactivation events can cooperate to induce malignant transformation, has remained elusive”.

Contrary to what has been previously reported [25], the artificial ectopic overexpression of a limited number of genes does not promote the neoplastic transformation of normal cells [26]. Furthermore, cancer transformation triggered by overexpression of oncogenes has been performed in animal models, whereas human cells display high resilience when tested, according to these artificial manipulations [27]. Indeed, mutated oncogenes or cellular growth regulatory genes, when incorporated into normal human epithelial cells, failed to immortalize or transform these cells [28].

Additionally, mathematical models have shown the inconsistency of experiments in which an increasing number of mutated genes participate, given that the expected effect is frequently small due to the redundancy of the genetic and regulatory pathways involved in shaping a specific phenotype [29]. Moreover, it is quite disturbing that several mutations have only been found in a limited percentage of cancer cells of the same tumor type, while a significant proportion of neoplastic cells do not show any mutated genes [30,31,32]. In several cases, mutations identified during the early stages of cancer development are lost during disease progression, indicating that they are not required for further cancer evolution [33,34,35,36]. These findings cast doubt on the relevance of “driver” mutations, given that a tumor can progress even without the help of such genetic changes. One might suppose that “initiation” and “progression” rely on very different mutations. In this case, targeting those genes that are presumed to “initiate” the transforming event would be useless for inhibiting cancer progression. Moreover, cancer genomes are unstable and undergo continuous remodeling, which ultimately confers a highly heterogenic profile to cancerous cells [37]. Nonetheless, cancers “retain” their malignant [38] and metastatic capability [39], even after spontaneous loss of the mutated gene(s). While a distinct set of genes might participate during different stages of cancer development, doubts about their role as the “cause” (i.e., causative agents) of the cancer are mounting. Thereby, the search for an “indispensable” mutated gene (or a set of modified genes) that would transfer its cancerous features to a cell/tissue is still open. As a proof of concept, genetic changes and/or mutations usually associated with cancer have also been found in non-cancerous tissues [40,41,42,43,44]. This result further undermines the concept of mutation as “cancer-specific”, given that at least some superimposable genetic changes are observed in non-cancerous tissues [45,46]. 

A clear example of the debatable relevance of mutation profiles in cancer comes from the systematic review carried out by [47], in which the author demonstrates, through a careful analysis of the literature, that of the 17,371 human genes appearing in at least one paper in PubMed, 15,233 (87.7%) are also appeared in at least one paper mentioning cancer. This impressive result tells us that, if practically any gene is associated with cancer, which is the most probable case, looking at single gene explanations is devoid of any heuristic power. 

A case in point is the burden of mutations associated with tobacco. The K-*ras* oncogene is generally thought to be the critical genomic target of the chemical components of smoking [48,49]. Among lung cancers, adenocarcinoma shows K-*ras* mutation in a high proportion of cases. At odds with these findings, adenocarcinoma shows the weakest epidemiological association with tobacco, as other histological cancer types (in which K-*ras* mutation is rarely found) are represented prevalently among smokers [50]. This result does not mean that tobacco does not play a causal role in lung cancer; instead, it means that tobacco induces non-adenocarcinoma lung tumors through non-mutational mechanisms [51]. As a proof of concept, during the last two decades we have registered a steady increase in lung adenocarcinoma incidence in Taiwanese population, while the smoking rate of the population was significantly reduced [52].

Further confusion comes from studies in which target-based drugs have been used to target the allegedly “causative” mechanism. For instance, Imatinib, a specific inhibitor of the BCL-ABL oncogene located within the Philadelphia chromosome, can efficiently inhibit leukemia progression. However, clinical results do not accommodate this premise, given that leukemia cells *without* this mutation in the Philadelphia chromosome show better growth in culture than cells bearing the mutation [53]. This result calls into the question the causative role of this oncogene in allowing unrestrained growth. Indeed, if the mutation targeted by Imatinib does not support cancer proliferation, how could the drug hamper leukemia progression? Likely, Imatinib exerts antitumor effects through different mechanisms, namely by modulating tumor metabolism or its relationship with the microenvironment [54,55]. Interestingly, both effects can be accommodated by TOFT, while they can hardly be explained by SMT [56]. Additionally, evidence that casts doubt on the “causative” role of mutations in the tyrosine kinase BCR-ABL gene has been provided by studies using tyrosine kinase inhibitors. This class of drug can obtain significant objective responses, yet without any benefit in terms of increased survival. This paradoxical result has been explained by recalling that a cluster of cancer stem cells continue to proliferate because they lack the mutation “targeted” by the drug, and are therefore unresponsive to Imatinib [57,58]. The conclusion towards which we are headed is absurd: the only cancerous cells that can proliferate upon treatment are precisely those that lack the “oncogene” from which the cancer is supposed be “generated”.

Additionally, as the gene-based paradigm posits that once a gene is mutated/modified, a consequent gain- or loss-of-function swiftly arises. Thereby, we should expect that upon treatment with a genotoxic carcinogen (and the following appearance of a mutation), cancer would develop over a short time scale. However, it is well known that an exceedingly long latent period between carcinogen exposure and cancer initiation is required [59]. How can this latency be explained if mutations are the “cause” of cancer? 

Finally, although modifications in a large number of target genes have been noticed in tumors, in “the vast majority of cases it has not been possible to determine whether a particular mutation initiated the process, or occurred subsequent to the development of uncontrolled growth.” [60]. Yet, this conundrum can be explained by the fact that rigorous criteria required to establish a causal link between mutation and cancer initiation are rarely satisfied [61]. These rules include that the alleged mutation should be specific for each carcinogen used, while other mechanisms should be discarded, just to assign the causative role solely and exclusively to the single mutated gene(s). Instead, it is conceivable that some kind of genetic changes arise because of either genetic instability or remodeling of the cytoskeleton [62,63,64]. 

To sum up, we may confidently conclude that mutations are associated with tumors, even if they might be irrelevant as a primary cause [65], and hence ‘‘are not likely to play a dominant part in cancer’’ [66]. As a consequence, many studies claiming a relationship between gene expression and clinical outcomes, “avoid to identify the primary endpoint of interest, or report effects on a cause-specific outcome” [67].

### 2.2. Non-Genotoxic Carcinogenic Agents

There are carcinogens that act through non-genotoxic mechanisms, i.e., without inducing any detectable point mutation, offer a puzzling paradox in carcinogenesis studies [68,69]. Several carcinogens are indeed deprived of mutagenic effects, strongly contradicting the well-known Ames’s axiom (“substances act as carcinogens because they have mutagenic activity”) [15]. Noticeably, this awareness was firmly established long ago, in the 1940s, when Berenblum observed that the correlation between mutagenic and carcinogenic activities “can no longer be used as strong support in favour of the somatic cell mutation theory of cancer” [70]. 

Non-genotoxic carcinogens include tumor promoters (1,4-dichlorobenzene), endocrine-modifiers (17β-estradiol and endocrine disruptors), receptor-mediators (2,3,7,8-tetrachlorodibenzo-p-dioxin), immunosuppressants (cyclosporine), or toxicant/inflammatory agents (metals, such as arsenic and beryllium) [71,72]. Unfortunately, such findings are usually overlooked, given that such data could hardly be explained within the framework of SMT.

Foreign-body-based (FBC) carcinogenesis [73], i.e., the development of tumors in the proximity of inserted materials, devoid of intrinsic chemical activity, constitutes a relevant example of non-genotoxic carcinogenesis. Furthermore, their clinical relevance has been gaining momentum, as the incidence of tumors arising as a consequence of artificial prostheses steadily increases [74,75]. It is intriguing that what matters most is the physical shape of the implant (rather than its chemical nature). For instance, polymeric plates exhibit a significant carcinogenic property when inserted as integer surfaces, whereas they completely lose carcinogenicity when introduced as small fragments [76,77]. Several attempts have been made to explain these uncanny results [78], but unfortunately, none of them could accommodate the theoretical premises of SMT. Consequently, interest in FBC has rapidly declined, to the point that this argument is rarely mentioned in textbooks. This attitude is unbecoming and unworthy of science. Noticeably, it is worthy of interest that TOFT may offer a testable and plausible explanation, given that the insertion of a foreign body may disrupt normal cell–microenvironment interactions and can promote local tissue disorganization [79].

### 2.3. Paradoxical Molecular Effects on Cell Signaling

Molecular biology strictly relies upon a paradigm borrowed from the information theory: biochemical molecules carry “specific” information and transmit a “signal” by recognizing “targets”, according to a “lock-and-key” model, as vindicated by the antibody–antigen reaction [80,81]. Undoubtedly, this model fit in several instances, although in other settings the interactions that resulted were a little bit more complicated. Indeed, this model implies that the “biological signal” must be unambiguous and efficiently transmitted, and able to dampen environmental noise. The adoption of concepts from the information theory has been widely criticized [82,83]. Namely, compelling evidence has shown that numerous “signaling molecules” lack specificity and may eventually trigger opposite “paradoxical” effects. For instance, AKT—a key mediator of several cell survival and proliferation pathways—is considered to play a pivotal role according to the SMT [84]. However, by using a different methodological approach, it has been shown that Akt activation may paradoxically lead to unexpected inhibition of cell motility and proliferation [85,86]. Similarly, in rats overexpressing the K-*ras* oncogene, the deletion of p53—a gene usually conceived as having tumor suppressor activities [87]—almost completely abolished the carcinogenic effects supposedly exerted by K-*ras* [88]. This paradoxical behavior also occurs with the overexpression of p53 in mice exposed to chemical carcinogens [89]. Interestingly, such a result cannot be ascribed to any specific “signal” triggered by the activated gene, but instead should be explained by the “genetic instability” enacted by the artificial manipulation of the genome. The latter observation can help us in understanding how, by perturbing the overall system configuration, one can enact the neoplastic transformation process.

The NF-kB pathway, which is often overexpressed in cancer, provides another useful example, as its inhibition is thought to hinder tumor development. However, the silencing of Ikkb—a kinase required for NF-kB activation—unexpectedly favors hepatocellular carcinoma progression triggered by chemical carcinogens [90]. No consistent explanation for this manifest conundrum has been proposed. As happens much too frequently, authors merely state that such a “contradiction underscores the complexity of hepatocarcinogenesis, and predicts uncertainty in targeting these molecules”. It seems to us that this is a circumstantial sentence with little meaning. Instead, these kind of results further stress that “signaling” inside cells cannot be considered a linear process, given that the final outcome is strongly constrained by the context, and may hence lead to opposite consequences.

In humans affected by chromosomal/genetic diseases (such as trisomy 21), the gain or loss of genetic function due to mutation or gene deletion is subject to a robust modulation effect applied by tissue constraints that can either enact or inhibit the activity of those genes that are “mutated” [91]. It is arguable that when cells are approaching a phase transition, and the weight of constraints becomes critical in controlling opposite issues (i.e., in driving the system towards “alternative” attractors), then some genes can act in a paradoxical way. These instances may explain why mutated genes are ineffective in resting tissues, and why the importance of some pathways (such as those influenced by p53) emerge as critical only during some specific developmental steps. Activation of these pathways may lead to opposite outcomes, depending either on the tissue context or on the specific phase of the cell cycle [92,93,94].

Further evidence of the paradoxical role played by some molecules comes from studies conducted on hyaluronan (HA) [95]. Deposition of HA (in either tumor stroma or in cancer cells) is a widely accepted indicator of poor prognosis [96]. However, higher tissue concentrations of hyaluronidase (the enzyme that degrades HA) resulted in more malignant cancers than in low aggressive tumors; moreover, HA positively correlates with tumor invasiveness [97]. Again, we are facing with a true paradox: increase or decrease in a molecular player both established a positive (significant) correlation with tumor malignancy. We assume that it is hard to extract a sound meaning from these kinds of studies. 

The behavior of leucine-rich repeats and immunoglobulin-like domains protein 1 (LRIG1)—representing an endogenous inhibitor of growth factor and an alleged tumor suppressor, respectively—offers a further example of an inextricable puzzle. High LRIG1 expression has been associated with poor survival in prostate cancer patients in a Swedish study [98]. On the contrary, among US patients, high LRIG1 expression was significantly and unexpectedly associated with long-term overall survival [99]. Noticeably, the authors of the latter study concluded that the activity of “signaling molecules” depends on the specific experimental context adopted, and that a systems biology approach is needed to grasp the true meaning of such results. Indeed, a reductionist model, given the inconsistency of classical cause-and-effect paradigms when we are facing complex systems, can offer no reliable explanation [100].

Further paradoxes have been recorded for other signaling molecules, including oncogenes or antiapoptotic factors. For instance, Bcl-2 protein exerts an anticarcinogenic effect when activated in transgenic mice overexpressing c-*Myc* [101]; in addition, Bcl-2 is usually thought to be a pro-survival factor. In turn, c-*Myc*, a widely recognized “oncogene”, can unexpectedly induce both differentiation and apoptosis in embryonic human cells [102]. Currently, that c-*Myc* can promote differentiation and/or apoptosis in some contexts is an established fact [103]; however, it still awaits a convincing explanation, given that it is hard to reconcile the aforementioned properties with its pro-carcinogenic activity [104]. It is worth noting that c-*Myc* effects seem to be fine-tuned in opposite ways, either by cell density [105], and/or the concomitant activation of the E2F1 pathway [106]. This is a clear example of how “signal transmission” inside the cell can undergo symmetry breaking upon the influence of environmental biophysical/biochemical constraints. Similar considerations apply to the contradictory role played by the lysyl oxidase enzyme, regulated by the LOX gene. Scientific evidence demonstrates that LOX can have both tumor suppressor and metastasis enhancer activities [103]. These controversial results have been ascribed, as usual, to context-related differences that can modulate LOX dynamics. Quantitative changes of a molecular player may be deprived of meaning if we fail to refer to the milieu in which cells are embedded, as constraints can exploit the increase in molecule availability by driving the systems toward opposite outcomes when a critical transition is approaching. Thereby, the complexity emerging from that intricacy “precludes traditional microarray-based research to investigate tumor suppressor/metastatic promoting functions of LOX in human cancers” [107].

The TGF-β-related pathway is among the best-investigated examples of molecular paradoxes in cancer [108]. Both anticancer as well as cancer-promoting effects have been attributed to TGF-β. Currently, it is recognized that the ambiguous nature of TGF-β critically depends on the tissue context [109]. These kinds of studies have further highlighted how important the stroma is in cancer initiation and progression, probably because several stroma components dramatically determine whether TGF-β suppresses or promotes tumor formation [110]. TGF-β effects may be opposite in the presence of different stiffness values within the same tissue; for instance, in floating extracellular matrices with low stiffness levels, TGF-β stimulated contraction directly as well as indirectly (by pre-activating cells to express the myofibroblast phenotype), whereas when cells were cultured on matrices with high stiffness, TGF-β had no significant effects on contraction [111]. This paradigmatic example is instrumental in demonstrating how—at least in some cases—contradictory results can be conveniently solved by enlarging the perspective and moving from a lower to a higher level of observation. 

## 3. A Testable Hypothesis

Overall, the data we discussed above suggest that gene function cannot be dissociated from the phenotypic configuration in which the genome activity is embedded. Usually, we refer to this feature as a “context-dependent” effect. Yet, this is an abused formula as long as a convincing mechanistic explanation can be advanced. We believe that no solution can be envisaged if we persist in viewing the cell as a homogeneous, isotropic milieu in which chemical interactions occur according to a linear dynamic. Instead, cells are heterogeneous entities characterized by complex architecture and topological organization that segregates individual molecular components, thus allowing chemical reactions to be compartmentalized [112]. Namely, both cytoskeleton and nucleoskeleton modifications play a relevant role in enabling/blocking a number of interactions, eventually leading to gene expression at different/opposite ends [113]. Let us now discuss the following model to highlight the complex interactions that are likely to unveil the intricate relationships occurring between gene activity, physiological functions, and cell “context”.

Normally, a cell population dwells in an attractor, located within a complex phenotypic state–space landscape, such as proposed by C. Waddington and further conceptualized by Kauffman and Huang [114], among others. The attractor’s boundaries are shaped by a set of internal/external constraints—including tensional forces provided by cell-to-cell and extracellular matrix (ECM) contact—that compete in modeling a specific Gene Regulatory Network (GRN). Cells undergoing cell differentiation and/or experiencing a phenotypic switch across that landscape are “jumping” from one attractor to another. This process resembles a first-order transition, and chiefly involves the architecture of both cytoskeleton and nucleoskeleton. Consequently, several topological rearrangements occur, which significantly modify the chromatin’s accessibility, the spatial distribution of structure-anchored substrates, and the intracellular distribution of enzymes, thus significantly affecting the overall chemical dynamics. According to our hypothesis (Figure 1, α), the activation of the gene X, with subsequent synthesis of the correlated protein X1, is thought to interact with the substrates A and B, leading to the resulting product C. Overall, within the attractor Sp1, the process assumes the form of a ”linear” equation:X → X1 → X1 + A + B → C(1)

This equation preserves its linearity if the system maintains its context invariance, i.e., if it still occupies the same attractor. However, downstream of a phase transition—such as that occurring during mitosis, differentiation, or phenotypic change—context cannot be considered constant anymore. Imagine that the system is undergoing a shift from the SP1 to the SP2 state, in which, for example, the system acquires mesenchymal features along the epithelial–mesenchymal transition (EMT). Within this new “context”, the X1 gene product cannot physically interact with the native substrates A and B (Figure 1, β). Instead, the gene X can now establish a relationship with the substrate Z, leading to the appearance of the “new” product R.
X → X1 → X1 + Z → R(2)

Ultimately, the activation of the same gene X enables the triggering of novel pathways, absent in the previous phenotypic context, while the product C, released in the previous setting, is actually silenced. Through this process, the activation of the very same gene can eventually lead to the promotion of different and even opposite pathways. Noticeably, we should stress that the switching between the two configurations (α, β) definitely depends on the attractor (the “context”) in which the gene activity occurs.

An illustrative example is offered by ARID1A, a chromatin-remodeling gene that is commonly mutated in cancer, and hypothesized to be tumor suppressive. Indeed, ARID1A is highly expressed in primary tumors, while, after initiation, loss of ARID1A in metastatic lesions accelerates tumor progression [115]. Similarly, Bone Morphogenetic Proteins (BMPs)—belonging to the TGF-β superfamily [116]—can play paradoxical and even opposite effects on cancer development and progression by serving as either tumor promoters or tumor suppressors, acting on both canonical and non-canonical pathways [117].

Indeed, different accessibility to chromatin could, in principle, support the actions of both oncogenic and tumor-suppressive networks, resulting in opposite effects. The net carcinogenic effect likely stems from the context in which the mutations are found, such as tissue architecture, cytoskeleton rearrangements, microenvironmental influences, cell density [104], timing, or dose [118]. Compelling evidence suggests that several oncogenes—including APC gene mutations [119], claudins [120], NKX2-1/TTF-1 [121], RUNX proteins [122], GATA3 X308 [123], Fyn-related kinase [124], and c-RAS [125]—can exert opposite effects, which are substantially different depending on the cellular setting. It is worth noting that such a context is actively “shaped” by cell–microenvironment interactions, which can modify the chromatin opening and the gene expression pattern by modulating the cytoskeleton–nucleoskeleton interplay [126]. 

Incidentally, this is no different from the main tenet of control theory in engineering “there is no direct input/output relation that is not mediated by the actual state of the system” (Lynch, Nancy A., and Mark R. Tuttle. An introduction to input/output automata. Laboratory for Computer Science, Massachusetts Institute of Technology, 1988.). It is somewhat sad to recognize how the progressive fragmentation of science caused such basic issues to be forgotten.

## 4. Conclusions

Controversial results are frequently encountered in cancer studies. They resemble “paradoxes” if we strive to search for an interpretation based upon the SMT framework. However, we are reaching a point of no return, given that the accumulated weight of such disturbing results threatens to drive carcinogenesis studies into a blind alley. Indeed, a recent survey highlights that the majority of high-impact studies associating genetic biomarkers with cancer outcomes did not define or identify the primary endpoint of interest, or report effects on a cause-specific outcome. Among these studies, less than half reported effects on a cancer-specific endpoint [67]. This methodology leaves open the possibility that observed associations are unrelated to cancer.

Such a statement implies that the reductionist-based framework, which has shaped the theoretical background of molecular biology [127], is inadequate for grasping the complexity embedded in the carcinogenic process. In fact, emergent phenomena concerning morphogenetic and neoplastic processes involve a plethora of interactions among different entrenched levels, in which the lower levels (molecules, cells) are strongly influenced by higher level organization constraints [18,128].

Furthermore, this systems biology-based approach explains how cancer can either undergo a spontaneous regression or can revert toward a normalized phenotype, when experimentally “forced” by molecular or physical constraints. The spontaneous disappearance/differentiation of animal and human cancers have been frequently reported in the scientific literature [129,130,131,132]. In addition, experimental studies have demonstrated that, by placing cancerous cells into a “normal” microenvironment, tumors may differentiate, eventually ending in tumor reversion. These findings have been recorded for cancer cells cultured in an embryonic milieu [133,134,135] or cultured in a 3D reconstructed biological microenvironment mimicking the normal tissue architecture [136,137,138,139]. In these models, cells undergo entrenched processes of apoptosis and differentiation, culminating into the reestablishment of a “normal” phenotype [140,141]. These results are frequently underappreciated, which is precisely why SMT cannot provide a cogent explanation. We can confidently argue that if cancer is “caused” by the mutation of a set of key driver genes, once a threshold has been crossed in the “accumulation” of such events, there would be no way back towards normality. However, experimental evidence stands against this hypothesis. In fact, tumor reversion, as unusual as it sounds, can more properly accommodate the theoretical framework provided by TOFT, which posits that the interplay between cells and their microenvironment may trigger differentiation processes, including the recovery of a normal phenotype by cancerous tissues [142]. 

Overall, such paradoxical data call for a reappraisal of the oncogenic paradigm. Ignoring these data is no longer tenable, as “troublesome” as they may be. Avoiding facing conundrums is tantamount to saying that the prevailing theory holds in all instances except the paradoxical ones, where the current paradigm is explicitly contradicted. Ignoring a paradox would prevent us from reaching outstanding achievements. Rigorous scientific investigation requires us to ask the right questions and to place it within the proper “context”. The context is constituted by the level of biological organization where the phenomenon we are investigating occurs. Accordingly, a new theoretical and methodological approach is required to grasp the complexity of cancer. To address such issues in a proper manner, we should rethink our basic methodology, shifting from a 2D- to a 3D-based approach. Even this will not be enough: 3D constructs should incorporate extracellular matrix, cells (fibroblasts) of the stroma, as well as endothelial cells. Furthermore, such models should be capable of reproducing different stiffness values, collagen densities, and compartmentalization among different cellular constituents [143]. Briefly, future developments within this field require that we move toward modeling the extracellular matrix in a physiological fashion. Specifically, careful attention needs to be paid in “reconstructing” the tumor stroma [144], given that tumors should to be grown in a physiologically relevant environment receiving the same cues as they would in the native tissue.

The use of organoid-based culture—eventually coupled to microfluidic devices—are likely to better reflect the architecture and cellular composition of a tumor [145]. The experimental simulation of this architecture would enable us to appreciate both the complexity and the dynamic interactions tacking place within a tumor. Fruitful alternatives have been proposed in the last two decades [146], but there is still a “long way to Tipperary” before we obtain a coherent, new theoretical framework.

## Figures and Tables

**Figure 1 biomolecules-12-00662-f001:**
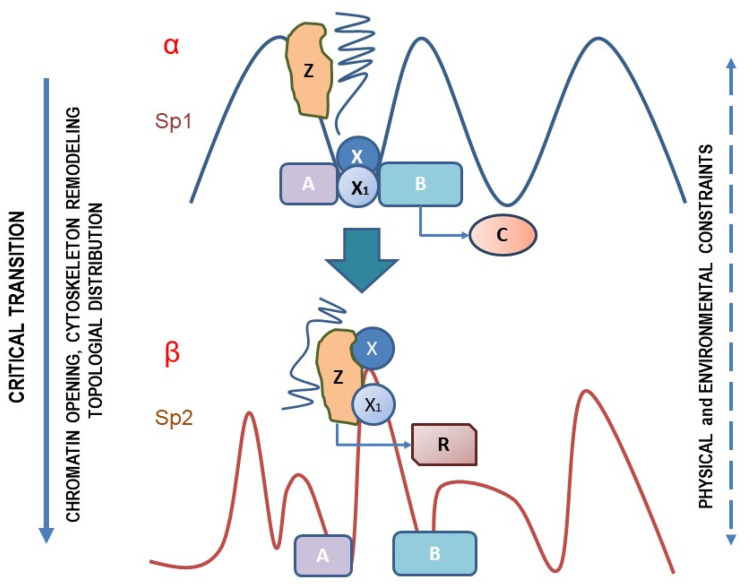
According to the hypothesis discussed in the text, the activation of the gene X, with subsequent synthesis of the correlated protein X1, is thought to interact with the substrates A and B, leading to the resulting product C. However, downstream of a phase transition the context cannot be considered constant anymore. When the system is undergoing a shift from the SP1 to the SP2 state, for instance the X1 gene product cannot physically interact with the native substrates A and B. Instead, the gene X can now establish a relationship with the substrate Z, leading to the appearance of the “new” product R. Ultimately, the activation of the same gene X triggers a novel pathways, absent in the previous phenotypic context, while the product C, released in the previous setting, is actually silenced. Ultimately, through this process, the activation of the very same gene can lead to the promotion of different and even opposite pathways.

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
