# Peer review of "Paradoxical Behavior of Oncogenes Undermines the Somatic Mutation Theory"

_biomolecules, 2022, doi:10.3390/biom12050662_

Round 1

Reviewer 1 Report

General comment:

The article by Monti et al is a reflection about the validity of the somatic mutation theory (SMT) in cancer based on oncogenes behaviors judged paradoxical by the authors.

This “provocative” writing raises a series of issues that question the SMT.

It is of interest to have a thoughtful view on theories that have guided cancer research in the last 50-70 years, but generalizations based on papers that over interpret the results may not always lead to fruitful conclusions. Therefore, I would recommend that the authors show some restraint or sobriety in their comments and deductions.

It’s difficult to give an analysis of all issues that are commented or criticized. This could be the subject of a book I will just give some examples.

Specific comments:

Paragrah lines 50-55

In a mixture of 2 independent ideas, it is considered that the rapid emergence of resistances to targeted therapies belongs to the demonstration that it is an argument against this SMT theory and that the rapid development of targeted therapies’ R&D is (only) guided by financial interest. I would surely agree with the last consideration and accept the idea that the industry could make cheaper treatments and keep significant profits.

However for the question of resistance to targeted therapies, I don't see how this contradicts the SMT. At least it should be discussed that tumor tissues in which mutations accumulate and generate a great cellular heterogeneity, the selection of resistant populations may happen rapidly.

Par lines 61-70

Authors: "Data from animal studies, especially those provided by inbred tumor-prone, tumor cell-injected, or knockout rodent models – carry a number of shortcomings and, in many cases, are mere artifacts, far from being reliable “models” "

I would not quote animal models as not reliable. Animal models giving clear-cut results are reliable, the question is are these models pertinent for the question that is addressed. The design of animal models is based on questions and hypotheses and this doesn't preclude that results should be carefully interpreted in view of the differences that exist between these models and human tumors. The authors say that the models are mostly designed to support the accepted theoretical assumptions, not for exploring divergent hypothesis or infer new perspectives. This assumption is too restrictive. Maybe they could give some examples in which they consider that conclusions are shortcomings based on biased interpretation of results.

Lines110-113

Authors "However, the demonstration of a mechanistic, truly causative link between mutations and cancer is still lacking, and the supposed correlation is suggestive for association and not causation."

I am not sure to understand this idea: is it against the oncogene theory of cancer or against the notion that there is a relation between the frequency of mutations and the level of tumorigenicity?

Lines 124-126

Mutations appearing early in the cancer course may disappear during the evolution of the tumor. I don't think that this an argument against the SMT. Several remarks can be made, for instance 1) all mutations are not equivalent and may not have the same importance, 2) aberrant expression of protein functions may lead to cell reprogramming without the necessity of permanent expression. This is seen in the reprogramming of somatic cells into iPS, 3) rewiring of cell signaling pathways may occur due to various events (including exogenous pressure) during the existence of a tumor .

Par Lines 138-144

Again a mutation observed in a tumor (cell) doesn't mean that this mutation is necessary to the tumor potential.

Line 148

"… adenocarcinoma shows the weakest epidemiological association with tobacco, as other histological cancer types – in which K-ras mutation is rarely found – are represented prevalently among smokers 42."

The reference is from 2004. I have not checked all the papers but to my knowledge the predominant type of lung cancer, NSCC, has largely shifted from squamous cell carcinoma to adenocarcinoma during the last 50 years and there is a large increase of women presenting this type of tumors, both likely because of changes in smoking behavior. Lung carcinoma remain strongly associated with tobacco.

I would be careful for the last sentence, "tobacco induces non-adenocarcinoma lung tumors through non-mutational mechanisms". Multiple mutations have been reported in fact (see for instance Surg Oncol Clin N Am 25 (2016) 447–468 http://dx.doi.org/10.1016/j.soc.2016.02.003).

Par Lines 246-255

Concerning the NF-kB pathway and contradictory results between hepatocellular carcinoma (induced by carcinogens) and other tumors, different tissues have different metabolism (and different environment as mentioned in the last sentence) and the malignant process may highjack opposite signaling pathways.

Line 256

Is trisomy 21 a good example of monogenic disease ?

Par Lines276-286

The refs 89 and 90 are cited in an appropriate manner. The authors should read carefully these 2 papers. Refs 89 Powell Cell is fundamental research on LRIG1 dealing with mechanisms of intestinal cancers mainly in animal models whereas Ref 90 Thomasson Int J Cancer is a clinical study relating expression of LRIG1 to clinical parameters in prostate cancer  in a Swedish and an American cohorts with different treatments.

Lines 287-298

Myc is an extremely complex actor in carcinogenesis and cell biology in general. It is difficult to give an overview of the problem and provide conclusions in 10 lines.

Par Lines 324-336

I think that these are general considerations with which a lot of people (maybe all) agree and many papers deal with these concepts. This has not to be presented as an original theory.

Par Lines 396-404

Could the authors give some examples of how the SMT framework may drive carcinogenesis studies into a blind alley. Does that mean that researchers are submitted to constraints that bias their research projects and impeach a more innovative and productive research to be developed.

Par Lines 411-429

The authors mention frequent reports of spontaneous disappearance of tumors. everibody would hope that it would be really frequent !!

In my experience, I don't think that such projects if well presented are rejected by scientific committees. On the other hand it might be true that article reviewers may require more controls, but this can be understood providing that they remain within a reasonable lab effort.   

Reviewer 2 Report

Issues of the Somatic Mutation Theory (SMT) were comprehensively reviewed. It covered that mutations cannot explain all causes of cancer, and functions of genes may change depending on the condition and situation. I agree that updated theoretical models for cancer will be useful for better diagnosis and treatment. Overall, this is a nice Opinion article.

  1. What will be the next-generation theoretical models for cancer? The manuscript clearly described that current models are too simplistic. I understand that cancer is a highly complex phenomenon. But, just saying “complex interaction between cells and their microenvironment” cannot be a model. More clear insights/opinions from the authors will be interesting.

Round 2

Reviewer 1 Report

The new version is an amended form of the previous article with responses and modifications aiming at responding to my comments that were only some examples of critics that I had concerning the multiplicity of issues that were addressed and some lack of sobriety in their interpretations.

Despite a substantial effort to improve the manuscript, the authors have only marginally changed the method.

On the other hand, they raise interesting questions that deserve to be discussed and they made a significant work to extract data from the literature to comfort their theories. I will certainly not oppose to the publication of such an article. The readers will make their own opinion.

By the way, the authors didn’t answer the following remark:

Par Lines276-286

The refs 89 and 90 are cited in an appropriate manner. The authors should read carefully these 2 papers. Refs 89 Powell Cell is fundamental research on LRIG1 dealing with mechanisms of intestinal cancers mainly in animal models whereas Ref 90 Thomasson Int J Cancer is a clinical study relating expression of LRIG1 to clinical parameters in prostate cancer  in a Swedish and an American cohorts with different treatments.